# Association of Problematic Alcohol Use and Food Insecurity among Homeless Men and Women

**DOI:** 10.3390/ijerph17103631

**Published:** 2020-05-21

**Authors:** Lorraine R. Reitzel, Surya Chinamuthevi, Sajeevika S. Daundasekara, Daphne C. Hernandez, Tzu-An Chen, Yashwant Harkara, Ezemenari M. Obasi, Darla E. Kendzor, Michael S. Businelle

**Affiliations:** 1Department of Psychological, Health & Learning Sciences, The University of Houston, 3657 Cullen Blvd Stephen Power Farish Hall, Houston, TX 77204, USA; sschinam@CougarNet.UH.EDU (S.C.); tchen3@Central.UH.EDU (T.-A.C.); yash.harkara@gmail.com (Y.H.); emobasi@Central.UH.EDU (E.M.O.); 2HEALTH Research Institute, The University of Houston, 4849 Calhoun Rd., Houston, TX 77204, USA; 3Department of Health & Human Performance, The University of Houston, 3875 Holman Street, Garrison Gymnasium, Room 104, Houston, TX 77204, USA; ssdaunda@Central.UH.EDU; 4Cizik School of Nursing, The University of Texas Health Science Center, Houston, TX 77030, USA; daphne.hernandez@uth.tmc.edu; 5Oklahoma Tobacco Research Center, The University of Oklahoma Health Sciences Center, 655 Research Parkway, Suite 400, Oklahoma City, OK 73104, USA; Darla-Kendzor@ouhsc.edu (D.E.K.); Michael-Businelle@ouhsc.edu (M.S.B.)

**Keywords:** alcohol use, homeless, drinking behaviors, alcohol dependence, food insecurity, sex

## Abstract

Food insecurity results from unreliable access to affordable and nutritious food. Homeless adults are particularly vulnerable to both food insecurity and problematic alcohol use. The current study examined the link between problematic alcohol use and food insecurity among homeless adults. Participants (N = 528; 62.7% men; M_age_ = 43.6 ± 12.2) were recruited from homeless-serving agencies in Oklahoma City. Problematic alcohol use was measured using the Alcohol Quantity and Frequency Questionnaire and the Patient Health Questionnaire. The latter used DSM-IV diagnostic criteria to assess probable alcohol use dependence/abuse. Heavy drinking was considered >7 drinks (women) and >14 drinks (men) per week. Food insecurity was measured with the USDA Food Security Scale-Short Form. The link between alcohol problems and food insecurity was examined with logistic regression analyses controlling for sex, age, education, income, and months homeless. Overall, 28.4% of the sample had probable alcohol dependence, 25% were heavy drinkers, and 78.4% were food insecure. Probable alcohol dependence and heavy drinking were correlated at 0.53 (*p* < 0.001). Results indicated that heavy drinking (OR = 2.12, CI_.95_ = 1.21, 3.73) and probable alcohol dependence/abuse (OR = 2.72, CI_.95_ = 1.55, 4.77) were each associated with increased odds of food insecurity. Food insecurity and problematic alcohol use are major issues among homeless populations; this study suggests they are associated. Future research is needed to shed light on potential causal mechanisms and on whether alcohol may take precedence over eating or food purchases.

## 1. Introduction

Approximately 553,000 people in the United States experience homelessness—the lack of a fixed, regular and nighttime residence—on any given night [1]. Individuals who are homeless face numerous challenges to maintain or improve their health, and as a result suffer disproportionate rates of morbidity and mortality [2,3,4]. Two prevalent health risk factors among homeless adults include food insecurity, which is the inability to acquire adequate nutritious food due to insufficient money or other means to enable sufficient food access [5,6], and alcohol use problems [7]. Although these constructs have been shown to be associated among domiciled samples [8,9,10,11,12], their association among homeless samples has not been extensively researched. 

The United States Department of Agriculture reported that, in 2018, 26.1 million domiciled adults (10.4%) were food insecure. There is no national prevalence rate for food insecurity among the homeless population, and the majority of extant research is limited to older adults, youth, and veterans. However, data suggest food insecurity ranges from 24% to 59.5% among these homeless subpopulations—rates much higher than among the domiciled population [13,14,15]. One study conducted among a more general sample of homeless adults indicated nearly ubiquitous food insecurity (i.e., 93.8%) [16]; however, this was an intervention study of a very small sample (N = 32). Reasons for the link between homelessness and food insecurity include a lack of stable income; reliance on others for food; lack of proper storage facilities for food; no kitchen, equipment, or utensils to prepare food; limited access to grocery stores to acquire food; and competing demands such as securing housing or income [17,18,19,20,21,22]. For individuals who are homeless, obtaining alcohol may also represent a competing demand to obtaining food given limited resources. 

Problematic alcohol use, including heavy alcohol use and alcohol dependency/abuse, are about 10 times more prevalent among homeless individuals than their domiciled counterparts [23,24,25]. Studies indicate alcohol dependence disorder rates of 8.1% to 58.5% [23], and at-risk drinking rates from 31% to 39% [7,26]. The lifetime prevalence of alcohol use disorders is about 60%, and the current prevalence rate is about 40%, among individuals who are homeless [27,28]. Among emergency department patients, those who were homeless reported greater levels of unhealthy alcohol use in the past year (44%); both current homelessness and experiences of homelessness within the past 12 months were significant predictors of unhealthy alcohol use [29]. The probable reasons for alcohol use problems among individuals experiencing homelessness include alcohol acting as a perceived agent for coping with psychological distress and/or physical pain, and the social advantage of having a sense of “community” around street culture and drinking [30]. Alcohol use problems and the myriad comorbidities that accompany them are strong contributors to the initiation and maintenance of homelessness, and have been associated with numerous adverse individual and societal consequences and high rates of morbidity and mortality among both domiciled and homeless adults [31,32,33,34].

Numerous cross-sectional studies have shown correlations between substance use and food insecurity (e.g., [35]), with some linking heavy alcohol drinking/alcohol dependence and food insecurity among domiciled samples [8,9,10]. Evidence suggests a bidirectional association between alcohol use and food insecurity. A majority of studies have identified food insecurity as a driving force for alcohol use problems [11,12]; however, some have cited alcohol drinking/alcohol dependence as a predictor of food insecurity [8,9,10]. No studies, however, were conducted with homeless individuals. 

Heavy alcohol use is linked with financial stress and unemployment, which are related to the risk of being food insecure [8,9,10,11,12,36]. Higher total alcohol consumption, binge drinking, and alcohol use disorders are also associated with unemployment [37,38]. Health problems due to heavy drinking are associated with lower chances of re-employment [39]. Unemployment leads to loss of income, which could increase the likelihood of experiencing food insecurity. In addition, heavy alcohol use may act as a barrier to obtaining adequate nutritious food regularly either by draining material resources [14] (spending their resources on alcohol rather than food), or by imposing a chaotic marginalized lifestyle that, in turn, predisposes individuals to food insecurity [14]. Therefore, alcohol use problems could be a significant predictor of food insecurity among homeless adults with limited resources and coping strategies.

The evidence is limited concerning the relation between alcohol use and food insecurity by sex. In general, substance use is more common among men [40,41], which may be due to greater impulsivity and higher rates of externalizing behaviors among men as a result of exposure to social stressors [42,43]. However, at least one study among homeless men and women indicated that women have equivalent rates of alcohol use problems relative to men [7]. Research among domiciled women indicates that they are more likely to be affected by food insecurity than men [44]. However, at least one study conducted among homeless adults indicated equivalent rates of food insecurity between the sexes [45]. Studies on the link between problematic alcohol use and food insecurity among domiciled groups also show conflicting results regarding whether the associations are predominately experienced by men or by women [9,10]. To date, no studies have evaluated the potential moderating effect of sex on the association between alcohol use problems and food insecurity among homeless adults.

The current study was designed to examine the link between problematic alcohol use and food insecurity among adults experiencing homelessness. Based on research evidence linking alcohol use and food insecurity among domiciled adults [9,10], it was hypothesized that heavy drinking and probable alcohol dependence/abuse would significantly predict food insecurity among adults experiencing homelessness. Based on studies that have displayed equivalent rates of alcohol use problems [7] and food insecurity [45] among men and women who experience homelessness, null moderation results were predicted. Understanding specific behavioral risk factors that predispose the homeless population to food insecurity is important for the success of interventions designed to reduce food insecurity among homeless adults. Examining associations by sex would further inform the currently mixed results in this area, and represent the first exploration of them among this disadvantaged population (homeless adults). 

## 2. Materials and Methods 

### 2.1. Participants

Participants were adults recruited from 6 homeless-serving agencies in Oklahoma City, OK, for a study on health and health behaviors using posted flyers. Individuals were eligible for participation if they were aged 18 years or older, demonstrated a >6th grade English literacy level on the Rapid Estimate of Adult Literacy in Medicine-Short Form (REALM-SF) [46], and received services (e.g., food, shelter, or counseling) at the targeted agencies. Overall, 610 individuals were enrolled in the study; however, 29 participants were further excluded from the analyses because they were not considered homeless based on responses to questions, “Where did you sleep last night” (i.e., selecting “My personal apartment or house”), “Are you currently homeless” (i.e., selecting “No”), current months homeless, and/or endorsing “I am not currently homeless” to the question “What are the reasons for your current homelessness”. 

### 2.2. Procedures

Data collection occurred on-site at each of the 6 agencies. Verbal informed consent was obtained for participation. Participants completed a computerized survey administered via tablet that enabled each study question to be read aloud. Recruitment and data collection spanned July and August of 2016. Participants received a $20 department store gift card for their time. 

### 2.3. Measures

#### 2.3.1. Sociodemographics 

Participants reported their sex, age, educational attainment, monthly income, and number of months spent homeless during their lifetime (“What is the total amount of time you have been homeless in your lifetime?”). 

#### 2.3.2. Problematic Alcohol Use 

Problematic alcohol use was assessed using 2 screening instruments. First, alcohol use frequencies and quantities were assessed using the Alcohol Quantity and Frequency Questionnaire [47], a self-report measure of average alcohol consumption on each day of the week over the last 30 days. Inquiries were accompanied by visual images of what constituted 1 standard drink (i.e., a 4–5-ounce glass of wine, a shot of liquor, a 12-ounce beer). Heavy drinking status was calculated from this measure and defined as consuming >14 drinks per week for men or >7 drinks per week for women over the last 30 days. This is commonly used as an alcohol screen in primary care settings [48,49]. This resulted in a binary (yes/no) outcome. “Probable Alcohol Dependence/Abuse” over the preceding 6 months was assessed using the Patient Health Questionnaire (PHQ), a 5-item self-administered screening instrument that uses diagnostic criteria from the DSM-IV [50]. This resulted in a binary (yes/no) outcome. Items included whether participants drank alcohol despite being advised by a doctor to cease for their health; drinking alcohol or being hungover while at work, school, or when taking care of other responsibilities; being late or missing work, school, or other events due to drinking or its effects; driving after drinking several drinks; and difficulty getting along with others due to drinking.

#### 2.3.3. Food Insecurity

Food insecurity was measured using a 6-item USDA Food Security Scale-Short Form [51]. The scale uses self-report to identify various aspects of food insecurity (e.g., not being able to afford to eat balanced meals; buying food that did not last and being unable to purchase more; going hungry due the inability to purchase food; cutting or skipping meals due to the inability to afford food, and the frequency with which this happened; and eating less than you should) over the preceding 12 months. Participants responding affirmatively to 0–1 of the items were categorized as food secure whereas those endorsing 2–6 of the items were categorized as being food insecure per the scoring convention [51]. 

### 2.4. Analytic Plan

Descriptive statistics, including correlations, were calculated for participant characteristics and other variables of interest. Two logistic regression analyses were used to examine the link between problematic alcohol use and food insecurity, each controlling for all sociodemographic variables. Estimates were expressed as odds ratios (OR) with 95% confidence intervals (CIs). An exploratory analysis was conducted to ascertain if sex was a significant moderator of the adjusted association between alcohol use problems and food insecurity, using an interaction term. The significance level was set at 0.05. All analyses were conducted using SAS, version 9.4 [52]. 

## 3. Results

### 3.1. Participant Characteristics

Overall, 9.12% (53/581) were excluded from analysis due to missing information on the variables of interest. Table 1 shows participant characteristics and correlations between study variables. The sample included in this study (*n* = 528) comprised 62.69% males (*n* = 331), 25% heavy drinkers (*n* = 132), 28.41% (*n* = 150) with probable histories of alcohol dependence/abuse, and 78.41% (*n* = 414) with food insecurity. Men were homeless for a longer lifetime duration than women (46.41 months vs. 34.44 months, *p* = 0.009), and had a higher prevalence of heavy drinkers (28.1% vs. 19.8%, *p* = 0.03) and probable alcohol dependence/abuse (33.23% vs. 20.30%, *p* = 0.001). However, the prevalence of food insecurity was not significantly different between men and women (78.25% vs. 78.68%, *p* = 0.907). Heavy drinking, probable alcohol dependence/abuse, and food insecurity were all significantly positively associated with one another. 

### 3.2. Main Analyses 

The adjusted logistic regression supported a significant association of problematic alcohol use and food insecurity. Specifically, heavy drinker participants had 2.12 (CI_.95_ = 1.21, 3.73) times higher odds of being food insecure than food secure. In addition, participants who had a history of probable alcohol abuse/dependence had 2.72 (CI_.95_ = 1.55, 4.77) times higher odds of being food insecure than those without a history of probable alcohol abuse/dependence.

### 3.3. Exploratory Analyses

Sex was not a significant moderator of heavy drinking (*p* = 0.96) or probable alcohol dependence (*p* = 0.32), respectively, or food insecurity, indicating that the association of heavy drinking and probable alcohol dependence with food insecurity did not differ between men and women. 

## 4. Discussion

Among this homeless sample, the risk of being food insecure was significantly higher for participants who were heavy drinkers or who had a history of probable alcohol abuse/dependence relative to their counterparts without alcohol use problems. Thus, results echo associations found between alcohol use problems and food insecurity in domiciled samples and extend findings to adults who are homeless [8,9,10]. Further, this study expands the literature on risky health behaviors that may contribute to food insecurity among adults who experience homelessness [53]. Unlike some prior research within domiciled groups comparing these associations by sex [9,10], however, the current study did not support that the link between alcohol use problems and food insecurity varies by sex. This was consistent with predictions, given the lack of sex differences for either health risk factor measured respectively within prior homeless studies [7,45]. Qualitative studies may help delineate if the basis of this link represents choices made to secure alcohol over food in the face of limited resources, as the current cross-sectional study cannot support causation. However, if true, results could suggest that interventions to affect problematic alcohol use among this group may have beneficial effects on food insecurity. More research is needed on this possibility. 

Findings of the current study indicated high rates of the health risk factors: overall, 28.4% of the sample had probable alcohol dependence, 25% were heavy drinkers, and 78.4% were food insecure. These rates may largely exceed those reported in domiciled samples [2,3,4,13,14,15,23] and suggest the need for access to empirically-based interventions to address these risks, which may contribute to high rates of morbidity and mortality seen within homeless studies. Unfortunately, prior studies indicate low rates of use of alcohol intervention among homeless adults (~17% in one study) [7]. This may be due to low readiness to change alcohol use behaviors, or a lack of access to ongoing care. Studies suggest, however, that between one-third to one-half of homeless adults want to change their drinking behaviors [26,54], lending evidence to low access being a primary issue. Likewise, research suggests that, although shelters and soup kitchens are common sources of meals for individuals who are homeless [17,55,56,57], meals offered in these settings may be nutritionally inadequate [17,56,57,58,59,60,61,62]. Additionally, meals may not be wholly accessible to all individuals if their availability is limited to certain times of day, which may conflict with appointments with others to address critical health needs and/or which may be impacted by health issues that prevent ambulation or travel to the site on time. Thus, systems-level attention is needed to address both alcohol use problems and food insecurity issues among individuals who are homeless. Promising approaches for addressing addictive behaviors among homeless adults include those implementing contingency management programs [63,64,65,66,67] and just-in-time smartphone-based interventions [68]; these may also represent promising avenues for future study in their effects on dietary behaviors among this group. 

Study limitations include those relevant to generalization of results given the use of a convenience sample from a single city who were English speaking and literate. However, it likely represented ~38.45% of the adult homeless population in Oklahoma City at the time of data collection [69]. Other limitations include the self-report of alcohol use problems, which may be subject to bias, and food insecurity, which may be affected by poor recall or inadequate nutritional literacy. Additionally, the food secure versus food insecure distinction scoring criterion is gross, and some items on the measure may be subjective in nature. Other food insecurity measures may provide a more detailed assessment of this construct, such as the Household Food Insecurity Access Scale [70], and could be used in future research. It should be noted that this study used screening instruments to assess problematic alcohol use that may have over- or underestimated true alcohol use disorders. Moreover, one of these measures used DSM-IV diagnostic criteria, which underwent changes in the DSM-5 to include the addition of craving [71]. Future studies might benefit from alternative or additional measures, including those using more current DSM-5 diagnostic criteria. Finally, the cross-sectional nature of this study precluded assumptions of a causal nature or of bidirectional associations. Relatedly, the different time periods for primary measures of interest (e.g., past 30 days, 6 months, and 12 months) further precludes causal assumptions. The inclusion of a sizable sample of homeless women, largely proportional to the percentage of homeless women in Oklahoma City [69], and the use of multiple recruitment sites were study strengths. Future work could extend exploration of differences by sex to differences by gender identity, and incorporate greater geographical representation. 

## 5. Conclusions

In summary, this study adds to the extant literature by supporting a link between problematic alcohol use and food insecurity among homeless adults that should be addressed in order to affect high rates of associated chronic disease and deaths among this vulnerable population. The association between problematic alcohol use and food insecurity did not vary by sex. Interventions designed to address coping strategies related to problematic alcohol use may also reduce food insecurity levels. 

## Figures and Tables

**Table 1 ijerph-17-03631-t001:** Sample descriptives and correlations between study variables (N = 528).

Study Variables	1	2	3	4	5	6	7	8	M (or N)	SD (or %)
**1. Sex (% Male)**	1	−0.12 **	0.04	−0.03	−0.11 **	0.09 *	0.14 **	−0.01	331	62.69
**2. Age**		1	0.11 **	−0.01	0.24 ***	−0.08	−0.05	0.01	43.59	12.11
**3. Years of Education**			1	−0.05	−0.04	0.07	−0.01	0.07	11.98	2.01
**4. Last Month Total Income**				1	−0.01	−0.05	−0.03	−0.07	364.49	642.88
**5. Months Homeless Lifetime**					1	−0.08	−0.08	−0.05	41.95	51.26
**6. Heavy Drinker** ^a^						1	0.53 ***	0.12 **	132	25.00
**7. Probable Alcohol Dependence/Abuse** ^b^							1	0.16 ***	150	28.41
**8. Food Insecurity** ^c^								1	414	78.41

*Note*. * *p* < 0.05; ** *p* < 0.01; *** *p* < 0.001; ^a^ reference group: non-heavy drinker; ^b^ reference group: no probable alcohol dependence/abuse; ^c^ reference group: food security.

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
