# Peer review of "Association of Problematic Alcohol Use and Food Insecurity among Homeless Men and Women"

_ijerph, 2020, doi:10.3390/ijerph17103631_

Round 1
Reviewer 1 Report
This article investigates whether or not there is a link between food insecurity and alcohol use in a homeless population in Oklahoma City. The authors found a relationship between food insecurity and problematic alcohol in homeless participants. This relationship was independent of sex, although higher drinking and homeless was found at a larger rate in males. The writing of the manuscript is clear and easy to read, which is much appreciated.
Identifying the needs of homeless populations is important to determining from what services they would benefit; however, a few points require clarification. Addressing these issues will hopefully strengthen the manuscript.
- Definition and prevalence of heavy drinking. The authors define heavy drinking as >14 drinks/week for men and >7 drinks/week for women. This is commonly used as an alcohol screen in primary care for at-risk drinking and does not rise to the level of a substance use disorder. While the CDC shares this heavy drinking definition, SAMHSA defines heavy alcohol use as binge drinking on 5 or more days in the past month (and binge drinking is 5 or more drinks per occasion within a couple hours of each other). A concern is that readers may overestimate the level of drinking of the participants in this study based upon the terminology versus screening criteria used. Even with these definitions/criteria used, the study found that 28% of men were “heavy drinkers” which is not far off from the 26% of adults in the general population that engaged in binge drinking in the past month (https://www.niaaa.nih.gov/publications/brochures-and-fact-sheets/alcohol-facts-and-statistics).
- Probable alcohol dependence/abuse. Why does this study use terminology and criteria from the DSM-IV instead of the DSM-5, which was updated in 2014? Alcohol dependence and abuse diagnoses have been replaced with alcohol use disorder, and the diagnostic criteria have been updated. Explaining how the screening questions that the authors used map onto current alcohol use disorder diagnostic criteria would be beneficial for the reader and relate easier to current research.
- Interpretation of relationship between drinking and food insecurity. This study found that that in their sample, 20-28% (women-men, respectively) were “heavy drinkers” and 20-33% had “probable alcohol dependence/abuse.” While these are clearly meaningful numbers, I am skeptical that these drinking behaviors are causing the food insecurity (78%) and homelessness that the authors state. And although the statistical tests show a relationship, they cannot demonstrate causation, so wording should carefully reflect this in the discussion. Additionally, I recommend comparing the results of problematic drinking found in this homeless population to that of the general public. For example, according to NIAAA, approximately 26% of adults engage in binge drinking within the past month, and 7% engage in heavy alcohol use (using the stricter definition explained in #1.). Given the prevalence of alcohol use and binge drinking in the general population, the prevalence of “heavy drinkers” in the homeless population sampled in this study is not striking, which raises the possibility that alcohol use is not driving homelessness and food insecurity. Without knowing about any other comorbid psychiatric disorders or medical illnesses and more about their lives, it is hard to interpret why participants are drinking.
- Sex differences. I commend the authors for examining sex differences in their study. This is a strength. It appears that participants reported whether they were male or female. This is fine; however, in the future including gender would be interesting, especially given the prevalence of homelessness and substance use disorders in transgender populations.
Author Response
Reviewer 1: Comments and Suggestions for Authors
Thank you for your positive comments and feedback regarding our manuscript.
Comment 1: Definition and prevalence of heavy drinking. The authors define heavy drinking as >14 drinks/week for men and >7 drinks/week for women. This is commonly used as an alcohol screen in primary care for at-risk drinking and does not rise to the level of a substance use disorder. While the CDC shares this heavy drinking definition, SAMHSA defines heavy alcohol use as binge drinking on 5 or more days in the past month (and binge drinking is 5 or more drinks per occasion within a couple hours of each other). A concern is that readers may overestimate the level of drinking of the participants in this study based upon the terminology versus screening criteria used. Even with these definitions/criteria used, the study found that 28% of men were “heavy drinkers” which is not far off from the 26% of adults in the general population that engaged in binge drinking in the past month (https://www.niaaa.nih.gov/publications/brochures-and-fact-sheets/alcohol-facts-and-statistics). Response: Thank you for this comment. In response, we have carefully gone through the manuscript and made clearer that we used two “screening instruments” to measure problematic alcohol use in this sample. Changes are made as follows:
- Methods (line 137): “Problematic alcohol use was assessed using 2 screening instruments.”
- Methods (line 143): “This is commonly used as an alcohol screen in primary care settings [48,49].”
- Discussion (lines 235-236): “It should be noted that this study used screening instruments to assess problematic alcohol use that may have over- or underestimated true alcohol use disorders.”
Comment 2: Probable alcohol dependence/abuse. Why does this study use terminology and criteria from the DSM-IV instead of the DSM-5, which was updated in 2014? Alcohol dependence and abuse diagnoses have been replaced with alcohol use disorder, and the diagnostic criteria have been updated. Explaining how the screening questions that the authors used map onto current alcohol use disorder diagnostic criteria would be beneficial for the reader and relate easier to current research. Response: Thank you for this very relevant comment. We used the Patient Health Questionnaire (PHQ) in this study, which used the more outdated DSM-IV criteria because it allowed comparisons of prevalence with some of our other homeless studies/samples in other states (not relevant to this manuscript) that were all conducted prior to 2013/4. It is important that we acknowledge the limitations of this in the present work, as well as comment on how this compares with the DSM-5 criteria. Thus, we add the following to the revised manuscript:
- Abstract (lines 22-24): “Problematic alcohol use was measured using the Alcohol Quantity and Frequency Questionnaire and Patient Health Questionnaire. The latter used DSM-IV diagnostic criteria to assess probable alcohol use dependence/abuse.”
- Discussion (lines 237-239): “Moreover, 1 of these measures used DSM-IV diagnostic criteria, which underwent changes in the DSM-5 to include the addition of craving [71]. Future studies might benefit from alternative or additional measures, including those using more current, DSM-5 diagnostic criteria.”
Comment 3: Interpretation of relationship between drinking and food insecurity. This study found that that in their sample, 20-28% (women-men, respectively) were “heavy drinkers” and 20-33% had “probable alcohol dependence/abuse.” While these are clearly meaningful numbers, I am skeptical that these drinking behaviors are causing the food insecurity (78%) and homelessness that the authors state. And although the statistical tests show a relationship, they cannot demonstrate causation, so wording should carefully reflect this in the discussion. Response: This is an important point and we now address it explicitly in the first paragraph of the Discussion (line 199, addition of ‘may’): “Further, this study expands the literature on risky health behaviors that may contribute to food insecurity among adults who experience homelessness [53].” Further, we add the following (line 205): “…as the current cross-sectional study cannot support causation.” Finally, in response to comments from Reviewer 2, we altered the Abstract conclusions (lines 31-35) to read: “Food insecurity and problematic alcohol use are major issues among homeless populations; this study suggests they are associated. Future research is needed to shed light on potential causal mechanisms and whether alcohol may take precedence over eating or food purchases.” Previous conclusions have been removed.
Comment 4: Additionally, I recommend comparing the results of problematic drinking found in this homeless population to that of the general public. For example, according to NIAAA, approximately 26% of adults engage in binge drinking within the past month, and 7% engage in heavy alcohol use (using the stricter definition explained in #1.). Given the prevalence of alcohol use and binge drinking in the general population, the prevalence of “heavy drinkers” in the homeless population sampled in this study is not striking, which raises the possibility that alcohol use is not driving homelessness and food insecurity. Without knowing about any other comorbid psychiatric disorders or medical illnesses and more about their lives, it is hard to interpret why participants are drinking. Response: Thank you for this comment and we agree on the usefulness of comparing domiciled and homeless studies on these points. We note, however, that a prior publication addressed the Reviewer’s request in greater depth (see Neisler et al., ref #7 in the list of citations, pp 44-45). Importantly, it used various manifestations of problematic alcohol use that went beyond the simple screening tools used in the present study and supported, for example, a greater prevalence of at-risk alcohol use (using the stricter definition) among homeless than domiciled samples. Moreover, as the Reviewer pointed out, there are some notable differences between the definitions of problematic alcohol use in the current study and the literature; these differences may preclude direct comparisons in this particular article that was relatively more strictly focused on alcohol use problems and food insecurity as opposed to alcohol use problem prevalence in this group. For example, the fact sheet referenced (https://www.niaaa.nih.gov/publications/brochures-and-fact-sheets/alcohol-facts-and-statistics) uses definitions that differ from those used in our study regarding how heavy drinking was defined and used terminology (and likely thus criteria) from the DSM-5 regarding “alcohol use disorder,” which differed from our measure of DSM-IV probable alcohol use/dependence. Thus, we believed the differences in definitions used between the face sheet and our work precluded direct one-on-one comparisons but to the Reviewer’s point added the following caveat to the Discussion (line 210, ‘may largely’): “These rates may largely exceed those reported in domiciled samples [2-4,13-15,23]...” Additionally, we believe that other modifications made to the manuscript as recommended by both Reviewers helps to highlight the nature of our instruments as screeners as opposed to modern diagnostic tools. This includes the following addition to the Methods: (line 137): “Problematic alcohol use was assessed using 2 screening instruments” and (line 143): “This is commonly used as an alcohol screen in primary care settings [48,49].” And the Discussion (lines 235- 242): “It should be noted that this study used screening instruments to assess problematic alcohol use that may have over- or underestimated true alcohol use disorders. Moreover, 1 of these measures used DSM-IV diagnostic criteria, which underwent changes in the DSM-5 to include the addition of craving [71]. Future studies might benefit from alternative or additional measures, including those using more current, DSM-5 diagnostic criteria. Finally, the cross-sectional nature of this study precluded assumptions of a causal nature or of bidirectional associations. Relatedly, the different time periods for primary measures of interest (e.g., past 30 days, 6 months, and 12 months) further precludes causal assumptions.” Thus, we sincerely hope that these changes, in combination of knowledge that another paper more directly and already addresses the Reviewer’s major point, satisfy the Reviewer’s concerns – particularly in combination with the other changes highlighted above in response to comment #3.
Comment 5: Sex differences. I commend the authors for examining sex differences in their study. This is a strength. It appears that participants reported whether they were male or female. This is fine; however, in the future including gender would be interesting, especially given the prevalence of homelessness and substance use disorders in transgender populations. Response: Thank you for this comment. We agree this would be interesting! In response to the Reviewer’s comment, we added the following to the revised manuscript (lines 244-246: “Future work could extend exploration of differences by sex to differences by gender identity…”
Reviewer 2 Report
Thank you for the opportunity to review the manuscript entitled “Association of Alcohol Use Problems and Food Insecurity among Homeless Men and Women” submitted to the International Journal of Environmental Research and Public Health. Overall this manuscript is well written and addresses an understudied and important line of research inquiry. My specific comments follow.
Abstract:
1- Criteria for probable alcohol dependence should be made clearer in abstract.
2- Interventions designed to address coping strategies related to problematic alcohol use may also reduce food insecurity. --> Despite the use of the word “may,” this statement lightly implies reasoning to believe causality, and your data are cross-sectional (as well as the items used rely on different time periods). I think the conclusion could be something to the effect of: food insecurity and alcohol consumption/dependence are major issues among homeless populations; this study suggests they are associated and future research is needed to shed light on potential causal mechanisms.
Background
1- Overall, well written. My only suggestion is that the interaction analysis seemed like a bit of an afterthought: “Similarly, understanding the role of sex in this relation could suggest and/or rule out the need for sex-specific interventions or programs that could better alleviate food insecurity among this disadvantaged population.” --> Given that this is cross-sectional data and therefore no information on whether interventions addressing alcohol use would affect food insecurity, there needs to be a more compelling rationale for the value of assessing this interaction effect. You wouldn’t withhold alcohol cessation or food access interventions from members of one sex because of any observed interaction effect here. I don’t know that this study needs this result to be important. If you keep this analysis, just improve justification for why you conducted in the background section.
Methods
1- Line 143: Unclear what is meant by “…was meant to capture true diagnostic cases…” Perhaps rephrase to improve clarity.
2- Should show all 6 food insecurity items asked. Also, why was 2 the cut-off? Was there a precedent set in the 1999 paper? Is this still applicable today? Also, you mention in discussion the limitation of alcohol items being self-report, but self-report not explicitly called out of the food insecurity item. For example, what is the nutritional literacy like for this group? Do they have a shared understanding of what a “balanced meal” is? Could be a potential study limitation.
Discussion
1- I think a notable limitation that should be discussed is the different time periods for primary measures of interest: past 30 days, 6 months, and 12 months. Thus, alcohol use/probable dependence/abuse may not have been at the same time as the experience of food insecurity. Overall this study still makes an important contribution to the research literature.
Author Response
Reviewer 2: Comments and Suggestions for Authors
Thank you for your positive comments and feedback regarding our manuscript.
Comment 1: Abstract: Criteria for probable alcohol dependence should be made clearer in abstract. Response: Thank you for this comment. In the revised manuscript, we expand on this measure in the Abstract (lines 22-24): “Problematic alcohol use was measured using the Alcohol Quantity and Frequency Questionnaire and Patient Health Questionnaire. The latter used DSM-IV diagnostic criteria to assess probable alcohol use dependence/abuse.” We had already expanded the abstract beyond recommended word imitations to address other Reviewer comments and thus did not expand further in this area. However, we added information in the Methods about this screening instrument to address the Reviewer’s comment (lines 146-151): “Items included whether participants drank alcohol despite being advised by a doctor to cease for their health; drinking alcohol or being hungover whole at work, school, or when taking care of other responsibilities; being late or missing work, school, or other events due to drinking or its effects; driving after drinking several drinks; and difficulty getting along with others due to drinking.”
Comment 2: Abstract: Interventions designed to address coping strategies related to problematic alcohol use may also reduce food insecurity. --> Despite the use of the word “may,” this statement lightly implies reasoning to believe causality, and your data are cross-sectional (as well as the items used rely on different time periods). I think the conclusion could be something to the effect of: food insecurity and alcohol consumption/dependence are major issues among homeless populations; this study suggests they are associated and future research is needed to shed light on potential causal mechanisms. Response: Thank you for this comment. We have amended the Abstract (lines 31-35) to read: “Food insecurity and problematic alcohol use are major issues among homeless populations; this study suggests they are associated. Future research is needed to shed light on potential causal mechanisms and whether alcohol may take precedence over eating or food purchases.” Previous conclusions have been removed.
Comment 3: Background: Overall, well written. My only suggestion is that the interaction analysis seemed like a bit of an afterthought: “Similarly, understanding the role of sex in this relation could suggest and/or rule out the need for sex-specific interventions or programs that could better alleviate food insecurity among this disadvantaged population.” --> Given that this is cross-sectional data and therefore no information on whether interventions addressing alcohol use would affect food insecurity, there needs to be a more compelling rationale for the value of assessing this interaction effect. You wouldn’t withhold alcohol cessation or food access interventions from members of one sex because of any observed interaction effect here. I don’t know that this study needs this result to be important. If you keep this analysis, just improve justification for why you conducted in the background section. Response: We maintained these analyses Because Reviewer 1 indicated it was a strength of our work. Accordingly, we amended the Introduction section to be clearer about the potential contribution of this exploration to the literature (lines 111-113): “Examining associations by sex would further inform the currently mixed results in this area, and represent the first exploration of them among this disadvantaged population (homeless adults). Further, we removed reference to any sex-specific intervention applications in the conclusions section (line 251).
Comment 4: Methods: Line 143: Unclear what is meant by “…was meant to capture true diagnostic cases…” Perhaps rephrase to improve clarity. Response: This phrase was removed from the revised manuscript.
Comment 5: Methods: Should show all 6 food insecurity items asked. Also, why was 2 the cut-off? Was there a precedent set in the 1999 paper? Is this still applicable today? Also, you mention in discussion the limitation of alcohol items being self-report, but self-report not explicitly called out of the food insecurity item. For example, what is the nutritional literacy like for this group? Do they have a shared understanding of what a “balanced meal” is? Could be a potential study limitation. Response: Thank you for all of these comments. Our responses are bulleted below:
- Should show all 6 food insecurity items asked. We have added all of the food insecurity items to the manuscript draft (lines 155-157). Now reads: “The scale uses self-report to identify various aspects of food insecurity (e.g., not being able to afford to eat balanced meals; buying food that did not last and being unable to purchase more; going hungry due the inability to purchase food; cutting or skipping meals due to the inability to afford food and the frequency with which this happened; eating less than you should) over the preceding 12 months.” Note that “cutting or skipping meals due to the inability to afford food and the frequency with which this happened” represents 2 discrete items, thus totaling 6 overall.
- Why the cut-off and what is the precedent. A score of 2 is the cut off per scoring convention (line 159). Now reads: “…whereas those endorsing 2-6 of the items were categorized as being food insecure per scoring convention [51].” That is the cut off that was established by the researchers at the National Center for Health Statistics in collaboration with Abt Associates Inc. This is documented in citation 51, setting the precedent for this scoring. This is also mentioned in a 2012 document, which can be accessed here: https://www.ers.usda.gov/media/8282/short2012.pdf, and which states For some reporting purposes, the food security status of households with raw score 0-1 is described as food secure and the two categories “low food security” and “very low food security” in combination are referred to as food insecure. Our coauthor (DCH) is an expert in food security and uses this scoring method in her work.
- Is the cut-off still applicable? The cut-off is still applicable today in that it is still widely used. In response to the Reviewer’s comment, we added (lines 232 – 233): “Additionally, the food secure versus food insecure distinction scoring criterion is gross, and some items on the measure may be subjective in nature.”
- Limitations of the measure with this sample. The Reviewer makes a good point about limitations. For example, eating a balanced meal can certainly be subjective and affected by nutritional literacy. However, not all items are subjective; for example, “skipping a meal”, “eating less because there was not enough money for food”, or not “eating (despite being hungry) because there was not enough money for food”, is less arbitrary. In response to the Reviewer’s points, we added (lines 230 – 233): “Other limitations include the self-report of alcohol use problems, which may be subject to bias, and food insecurity, which may be affected by poor recall or inadequate nutritional literacy. Additionally, the food secure versus food insecure distinction scoring criterion is gross, and some items on the measure may be subjective in nature.”
Comment 6: Discussion: I think a notable limitation that should be discussed is the different time periods for primary measures of interest: past 30 days, 6 months, and 12 months. Thus, alcohol use/probable dependence/abuse may not have been at the same time as the experience of food insecurity. Overall this study still makes an important contribution to the research literature. Response: The Reviewer makes a good point. This has been accounted for in the revised manuscript as follows (lines 241-242); “Relatedly, the different time periods for primary measures of interest (e.g., past 30 days, 6 months, and 12 months) further precludes causal assumptions.”